# Enhanced Biodegradation of Phthalic Acid Esters’ Derivatives by Plasticizer-Degrading Bacteria (*Burkholderia cepacia*, *Archaeoglobus fulgidus*, *Pseudomonas aeruginosa*) Using a Correction 3D-QSAR Model

**DOI:** 10.3390/ijerph17155299

**Published:** 2020-07-23

**Authors:** Haigang Zhang, Chengji Zhao, Hui Na

**Affiliations:** Alan G. MacDiarmid Institute, College of Chemistry, Jilin University, Changchun 130012, China; zhaochengji@jlu.edu.cn (C.Z.); huina@jlu.edu.cn (H.N.)

**Keywords:** diethyl phthalate, plasticizer-degrading bacteria, biodegradation, molecular modification, molecular dynamics

## Abstract

A phthalic acid ester’s (PAEs) comprehensive biodegradability three-dimensional structure-activity relationship (3D-QSAR) model was established, to design environmentally friendly PAE derivatives, which could be simultaneously degraded by plasticizer-degrading bacteria, such as *Burkholderia cepacia*, *Archaeoglobus fulgidus*, and *Pseudomonas aeruginosa*. Only three derivatives of diethyl phthalate (DEP (DEP-27, DEP-28 and DEP-29)) were suited for their functionality and environmental friendliness, which had an improved stability in the environment and improved the characteristics (bio-toxicity, bioaccumulation, persistence, and long-range migration) of the persistent organic pollutants (POPs). The simulation inference of the microbial degradation path before and after DEP modification and the calculation of the reaction energy barrier exhibited the energy barrier for degradation being reduced after DEP modification and was consistent with the increased ratio of comprehensive biodegradability. This confirmed the effectiveness of the comparative molecular similarity index analysis (CoMSIA) model of the PAE’s comprehensive biodegradability. In addition, a molecular dynamics simulation revealed that the binding of the DEP-29 derivative with the three plasticizer-degradation enzymes increased significantly. DEP-29 could be used as a methyl phthalate derivative that synergistically degrades with microplastics, providing directional selection and theoretical designing for plasticizer replacement.

## 1. Introduction

In industrial production, a plasticizer is an indispensable part of microplastics and increases the flexibility and durability of plastic products. The phthalic acid esters (PAEs) are the most widely used plasticizers [1]. About 8.4 million tons of PAE plasticizers are used annually, accounting for 70% of its total use worldwide. Among these, diethyl phthalate (DEP) accounts for a high proportion [2]. The extensive use of phthalate plasticizers creates great commercial value, though it also poses environmental health risks that we cannot ignore. PAE plasticizers are not directly connected to plastic polymers, and they easily release into the environment during use [3]. Agricultural plastic film is one of the primary sources of microplastics in the soil. The PAE content in a vegetable greenhouse base could reach up to 9.68 mg/kg [4]. The PAEs in microplastics belong to refractory organic compounds, which are detected in soil [5], rivers [6], drinking water [7], food [8], household garbage [9], sewage sludge [10], industrial wastewater [11], marine sediment [12] and landfill leach [13]. Toxicological studies proved that PAE plasticizers remaining in the environment could enter human and animal bodies through inhalation, diet and skin contact, causing great harm to human health and environmental safety [14]. Although PAEs are not acutely toxic to organisms, exposure to large doses could lead to teratogenic, carcinogenic and mutagenic effects in animals [15]. In addition, animal experiments confirmed that some phthalates cause liver and kidney damage in animals [16]. The US Environmental Protection Agency, the European Union, and the China National Environmental Monitoring Center have listed PAEs as the priority pollutants [17]. The degradation and transformation of phthalate plasticizers in environmental residual has become a topic of interest in recent years.

Microbial degradation is a key process that mineralizes the organic pollutants in the environment. Hoellein et al. [18] studied the biological effects of microplastics in the water surrounding sewage treatment plants. They observed that only specific microorganisms could affect migration behavior and the biodegradability of plasticizer. During microbial degradation, the PAEs could simultaneously be degraded by a variety of plasticizer-degrading bacteria. A study reported that 50% of the PAEs degraded after an inoculation of aerobic microorganisms in sewage sludge for 28 days [19]. Sugatt et al. [20] studied the biodegradability of 14 PAE plasticizers commercially used in microplastics using a shake flask test. They detected the PAE plasticizers were sensitive to the mixed microbial community in the natural environment, and they gradually degraded and mineralized. To improve the microbiological degradation efficiency of plasticizers, several scientists attempted to isolate a single species of bacteria with a high-efficiency of degradation from nature. Wu et al. [21] screened and isolated *Ochrobactrum lupini* and *Agrobacterium tumefaciens* that used dibutyl phthalate (DBP) as their only carbon source and energy source from polluted river sludge and completely mineralized it. The degrading enzymes (esterase, carboxylesterase and hydratase) play a crucial role in the synergistic degradation of PAE plasticizers in microplastics [22]. Under different environmental conditions, microorganisms degrade the PAE plasticizers, but the degradation cycle is relatively long [23]. Therefore, it is of immense significance to improve the biodegradability of the PAE plasticizers to inhibit their adverse effects on the environment and human health.

In this paper, we employed the range normalization method combined with the entropy weight method to characterize the comprehensive biodegradability of PAEs by enzymes from three plasticizer-degrading bacteria [24]. We constructed a three-dimensional structure-activity relationship (3D-QSAR) model of the comprehensive biodegradability of the PAEs. DEP was taken as the target molecule for molecular modification, and we screened for DEP derivatives suitable for degradation by plasticizer-degrading bacteria. Further, we evaluated the DEP derivatives for their functionality and environmental friendliness, providing directional selection for plasticizer replacement.

## 2. Materials and Methods

### 2.1. Data Source

The *Burkholderia*, *Archaeoglobus* [22] and *Pseudomonas* [25] species are typical plasticizer-degrading bacteria that effectively degrade PAE plasticizers. We obtained the structures of phthalate dioxygenase reductase from *Burkholderia cepacia* (PDB ID: 2PIA), esterase from *Archaeoglobus fulgidus* (PDB ID: 2ZYI) and carboxylesterase from *Pseudomonas aeruginosa* (PDB ID: 3CN7) from the Protein Data Bank (PDB (http://www.rcsb.org/pdb)) [26]. Using the molecular docking module in Sybyl-x2.0 software, we docked 17 different PAEs with the three degrading enzymes to obtain the corresponding scoring function value [27], which formed the source of the degradability data of the PAEs.

### 2.2. Construction of the Comparative Molecular Similarity Index Analysis (CoMSIA) Model of the Comprehensive Biodegradability of PAEs

We employed the range normalization method to standardize the degradation data of the three plasticizer-degrading bacteria and the entropy weight method to objectively weigh the standardized data [28]. Finally, we obtained the value of the comprehensive biodegradability of the PAE molecules, degraded by the three bacteria. The following are the range normalization formulas of the scoring function values of 17 PAE molecules after docking with the three plasticizer-degrading enzymes:
(1)$$Y_{ij}=\left(1-a\right)+a\frac{X_{ij}-X_{minj}}{X_{maxj}-X_{minj}}$$
(2)$$Y_{ij}=\left(1-a\right)+a\frac{X_{maxj}-X_{ij}}{X_{maxj}-X_{minj}}$$

Equation (1) is the processing formula for the positive index, where a higher index is desirable. Equation (2) is the processing formula for the negative index, where a lower index is desirable. Among them, *i* represents the different PAEs (*i* = 1, 2, 3, …, 17); *j* represents the degradability index of the different degradative enzymes (*j* = 1, 2, 3; corresponding to the enzymes from *B. cepacia*, *P. aeruginosa* and *A. fulgidus*, respectively); *Y_ij_* is the standardized value of each scoring function; *X_ij_* is the scoring function value of the *j*-th biodegradability index of the *i*-th PAE; *X_maxj_* and *X_minj_* are the maximum and minimum values of the corresponding indexes, a ∈ (0,1), which are generally 0.9. The scoring function value characterizes the index of microbial degradability with a higher score being desirable. Therefore, we used the processing formula for the positive index to standardize the microbial degradability data. 

The entropy weight method is an objective method for weight processing [29], which uses information entropy for processing the weight. The calculation steps are as follows:

Step 1: calculate the information entropy (*E_j_*) of the *j*-th biodegradability index of the *i*-th PAE using the following formula:(3)$$E_{j}=-k\cdot \overset{m}{\underset{i=1}{\sum }}Y_{ij}\cdot \ln Y_{ij}$$
where *k* = 1/lnm, m is the total number of PAE molecules (i.e., 17), and *E_j_* is the information entropy of the *j*-th biodegradability index.

Step 2: calculate the difference coefficient (*H_j_*) of the *j*-th biodegradability index, reflecting the difference in the degree of each index using the following formula:(4)$$H_{j}=1-E_{j}$$
where *H_j_* is the difference coefficient of the *j*-th biodegradability index.

Step 3: normalize the index difference coefficient (*H_j_*) to estimate the weight (*W_j_*) of each biodegradability index using the following formula:(5)Wj=Hj∑j=1nHj
where *W_j_* is the weight of the *j*-th biodegradability index, and *n* is the total number of biodegradability indicators (i.e., 3).

Step 4: calculate the comprehensive value of the biodegradability of the three plasticizer-degrading bacteria for each PAE using the following formula:(6)$$Z_{in}=\overset{n}{\underset{j=1}{\sum }}W_{j}\cdot Y_{ij}$$
where *Z_in_* is the comprehensive value of the biodegradability of the three plasticizer-degrading bacteria of the *i*-th PAE, and n is the total number of biodegradability indicators (i.e., 3).

The molecular structure of the PAEs and the comprehensive biodegradability CoMSIA model of the three plasticizer-degrading bacteria were mainly constructed by using the 3D-QSAR module in Sybyl-x 2.0 software (Tripos, Princeton, NJ, USA). The minimize module in the software optimized the molecular structure of the PAEs [30].To obtain a stable conformation of the lowest energy state of the molecule, the Powell’s conjugate gradient method was employed and combined with the Tripos molecular force field. The latter sets the charge of the molecule to the Gasterger–Hückle charge [31]. The energy convergence standard was adjusted to 0.005 kcal mol^−1^, and the number of iterations was set to 10,000 times [32]. In this paper, the diundecyl phthalate (DUP) with the highest comprehensive value of the biodegradability index of the three plasticizer-degrading bacteria was used as the template molecule (Figure 1). The Align Database module of the software was selected to select the common part of the optimized PAE molecular structure as the common skeleton for superposition [33].

Sixteen molecules were randomly selected from the comprehensive biodegradation values of 17 PAEs molecules and three plasticizer degrading bacteria. According to the ratio of 4:1, 13 PAE molecules were used as the training set, and the remaining three PAE molecules were used as the test set. The template molecule DUP was used in both the test set and the training set. The least squares method (PLS) was used to analyze the training set compounds. The training set was cross-validated under the extraction method module to obtain the cross-validation coefficient (q^2^) and the optimal number of principal components (*n*). The training set compounds were analyzed using the no validation regression to obtain the cross-validation coefficient (R^2^), standard deviation (SEE), test value (F) and the contribution rate of the force field (stereo field, electrostatic field). The scrambling stability test assessed the robustness of the model by evaluating the scrambling stability test parameters (Q^2^), cross-validated standard error of prediction (cSDEP), and dQ^2^/dr^2^yy. The cross-validation method assessed the external prediction ability of the built model to the test set compounds with the external verification parameter (r^2^_pred_). Based on the information prompted by the contour maps of the constructed CoMSIA model, the substitution sites and substitution groups that had a greater impact on the comprehensive score value were screened, forming the theoretical basis for the subsequent molecular modification. In this paper, DEP was used as the target molecule for modification, which is the most widely used and frequently detected in the environment. Figure 2 displays the molecular structure of the target molecule DEP.

### 2.3. Evaluation of Functionality and Environmental Friendliness of DEP Derivatives Based on Density Functional Theory (DFT)

Stability was used as an evaluation index for the molecular functionality of the DEP derivatives [34] with the frequency, total energy and energy gap as the parameters. The above parameters were calculated by Gaussian 2009 software (Gaussian Inc., Wallingford, CT, USA) based on density functional theory (DFT) at the unit level of b3pw91/6-31G* [35]; environmental friendliness was evaluated by the four persistent organic pollutants’ (POPs) characteristics of toxicity (LC_50_), persistence (logt_1/2_), bioconcentration (logBCF) and migration (−logPL) [36].

## 3. Results and Discussion

### 3.1. Construction and Evaluation of the CoMSIA Model for the Comprehensive Biodegradability of the PAEs by Plasticizer-Degrading Bacteria

#### 3.1.1. Calculation of the Comprehensive Biodegradation Values of the PAEs by Three Plasticizer-Degrading Bacteria of PAE Molecules

The docking score values, range normalized conversion values, weight and comprehensive biodegradation values of the combination of PAE molecules with phthalate dioxygenase reductase from *B. cepacia* (PDB ID: 2PIA), *A. fulgidus* (PDB ID: 2ZYI) and *P. aeruginosa* (PDB ID: 3CN7) are listed in Table 1.

#### 3.1.2. Construction of the CoMSIA Model for the Comprehensive Biodegradability of the PAE Molecules by Three Plasticizer-Degrading Bacteria

DUP exhibited the highest comprehensive biodegradability among all the PAEs. It was the template molecule for the common skeleton superpositioning of the training set compounds to construct a CoMSIA model of the comprehensive biodegradability of the PAEs by the three plasticizer-degrading bacteria. In Table 1, a and b represent the training set and the test set of the model, respectively. Table 2 lists the evaluation parameters of the CoMSIA model. The cross-validation coefficient (q^2^), calculated by the cross-validation analysis, was 0.731 (>0.5), and the principal component number (*n*) of the model was 8. Both of these indicate that the CoMSIA model had good internal prediction capabilities [37]. The standard deviation (SEE) calculated by the non-cross-validation analysis was 0.013 (<0.95), the test value (F) was 578.738, and the non-cross-validation coefficient (R^2^) was 0.999 (>0.9). These indicate that the model had a high fitting ability [38]. The perturbation stability test parameter (Q^2^) was 0.85, cSDEP was 0.251, and dq^2^/dr^2^yy was 1.488, indicating that the model had a high stability [39]. The external validation coefficient (r^2^_pred_), estimated by the external validation analysis of the test set compound, was 0.761 (>0.6), indicating that the model exhibited a good external prediction ability [40]. In addition, the contribution rate of each force field was 34.7% in the steric field (S), 11.1% in the electrostatic field (E), 46.8% in the hydrophobic field (H), 7.4% in the hydrogen bond acceptor field (D) and 0.0% in the hydrogen bond donor field (A).

#### 3.1.3. Contour Map Analysis of the CoMSIA Model 

The contribution rates of the steric, electrostatic field and hydrophobic field were the highest. Therefore, the force field information of the electrostatic and hydrophobic fields in the contour maps of the comprehensive biodegradability of the DEP molecule was analyzed (Figure 3). The steric field information map (Figure 3a) displays that introducing small-volume groups into the yellow region could improve the comprehensive biodegradation value of the DEP molecules by the three plasticizer-degrading bacteria. In other words, the groups with volumes less than that of -CH_2_CH_3_ were introduced at the C_1_ position. In the electrostatic field (Figure 3b), introducing positively charged groups into the blue region or negatively charged groups into the red region could effectively improve the comprehensive biodegradability of the DEP molecules [41]. However, the red region was located in the common skeleton and was difficult to replace and modify. Therefore, a group with a positive charge greater than -H introduced into the C_1_ position could achieve the purpose of increasing the comprehensive biodegradability. In the hydrophobic field (Figure 3c), introducing strong hydrophilic substituents in the white area at the C_2_ position was conducive to increase the comprehensive biodegradability of the DEP molecules. In summary, to improve the comprehensive biodegradability of DEP, it was necessary to modify its structure by introducing single and double substitutions: (i) at the C1 position, a group with a volume smaller than -CH_2_CH_3_ and a group with an electropositivity greater than -H, and (ii) at the C_2_ position, a hydrophilic group.

### 3.2. Molecular Modification of DEP for Enhanced Biodegradability Based on the CoMSIA Model

#### 3.2.1. Molecular Modification and Prediction of Comprehensive Biodegradability of DEP

According to the force field information from the contour maps of the CoMSIA model, the groups was substituted at the C_1_ and C_2_ positions to conduct single and double substitutions on the DEP molecule. The groups with volumes smaller than -CH_2_CH_3_ (-CH_3_, -OH, -H, -CN, -NH_2_, -CHO) and groups with an electropositivity greater than -H (-CH_3_, -CH_2_CH_3_, -CH(CH_3_)_2_, -C(CH_3_)_3_) were introduced at the C_1_ position. At the C_2_ position, the groups with higher hydrophilicities (-OH, -CHO, -COOH, -NH_2_, -COCH_3_, -CONH_2_) were introduced. These substitutions would improve the comprehensive biodegradability of derivative molecules. Thus, a total of 30 kinds of DEP derivatives were designed accordingly. Using the constructed CoMSIA model to predict the comprehensive biological properties of the modified derivatives, it was found that the comprehensive biodegradability of most of the DEP derivatives enhanced significantly (Table 3). Among them, the comprehensive biodegradability of 12 DEP derivatives increased by more than 15%, and the comprehensive biodegradation value of DEP-26 molecules had the highest growth rate, reaching 50.37%. DEP-23 molecules had the same substitution groups at site C_2_ as in the DEP-26 molecules. However, the H atom at site C_1_ (H_1_ atom) in DEP-26 resulted in a higher comprehensive biodegradation value than the corresponding substituent in DEP-23. DEP-21 was introduced to more hydrophilic groups, increasing its comprehensive biodegradability value to more than that of DEP-15. In summary, the prediction of the comprehensive biodegradability of the DEP derivatives by the three plasticizer-degrading bacteria was consistent with the information presented in the contour maps of the CoMSIA model.

#### 3.2.2. Verification of the CoMSIA Model for the Comprehensive Biodegradability of PAEs Molecules

In this paper, Sybyl-x2.0 software was also used to construct the 3D-QSAR model of the single biodegradability of the PAEs by the three plasticizer-degrading bacteria (Table 4). The DEP derivative molecules were screened through the biodegradable 3D-QSAR model of the PAEs’ molecular plasticizer-degrading bacteria, whose degradation efficiency ratio was close to the weight ratios of the entropy weight method in the comprehensive biodegradability model (Table 3). The docking score of the DEP derivative with phthalate dioxygenase reductase was 4.939–6.725, with a growth rate of up to 20.74%. Compared with the growth rate before modification, the docking score values of the DEP derivative binding with esterase ranged from −2.46% to 27.71%. The scoring functions for carboxylesterase and DEP derivatives exhibited a significant increase, with the growth rate ranging from 15.86% to 24.96%. The ratio of the increase in the biodegradation effect value of DEP-23–DEP-30 by three plasticizer-degrading bacteria was the closest to that calculated and weighted by the entropy weight method. These results verified that the CoMSIA model effectively provides the biodegradability information of DEP derivatives when degraded by three plasticizer-degrading bacteria. Additionally, this result demonstrated that the CoMSIA model modified by the entropy weight method is reliable with a good predictive ability, and can be applied to the design modifications in PAE molecules.

#### 3.2.3. Evaluation of the Functionality and Environmental Friendliness of DEP Derivatives

Molecular stability was the primary index for evaluating the functionality of DEP derivatives, characterized by the energy value, energy gap value, and positive frequency value. The persistent organic pollutant (POP) characteristics determined the environmental friendliness of the DEP derivatives, before and after modification, by evaluating their bio-toxicity (logLC_50_), bioaccumulation (logBCF), persistence (logt_1/2_) and long-range migration (log*K*_OA_) [36]. The energy value was inversely proportional to the stability of the molecules in the environment [42]. A higher positive frequency value indicated that the DEP derivative molecules could exist in the environment [43]. The bio-toxicity and long-range migration of the DEP derivatives were inversely proportional to the predicted logLC_50_ and log*K*_OA_ values, respectively. On the contrary, the bioaccumulation and persistence of the DEP derivatives were directly proportional to the predicted logBCF and logt_1/2_ values, respectively. Table 5 summarizes the predicted values of the parameters evaluating the functionality and environmental friendliness of the DEP derivatives, before and after modification.

Table 5 displays that the energy values of the DEP derivatives were significantly lower than that of the DEP molecule, with a reduction of 19.91–37.39%. The energy gap values did not change significantly before and after modification, indicating that the modified DEP derivatives were highly stable. In addition, the positive frequency values of the DEP derivatives were greater than zero, indicating that the modified molecules could exist in the environment. Among the eight DEP derivatives, only DEP-27, DEP-28, and DEP-29 exhibited no significant changes in their bio-toxicity compared with the unmodified molecule. The bio-toxicity of the remaining five derivatives significantly increased, with the highest increase-rate going up to 38.91%. An organic compound with a logBCF value less than 100 does not easily accumulate in organisms and has a small impact on the environment [44]. Except for the DEP-25 and DEP-26 molecules, the logBCF values of the DEP derivatives were much lower than 100. Among them, the logBCF values of DEP-27, DEP-28, and DEP-29 after modification did not change significantly. The predicted half-lives of the DEP derivatives were slightly higher than that of DEP, indicating that the degradability of these derivatives improved marginally. The predicted log*K*_OA_ values of the eight DEP derivatives were significantly higher than that of DEP, suggesting that their mobilities were decreasing. In conclusion, among the eight DEP derivatives, only DEP-27, DEP-28 and DEP-29 exhibited significantly higher biodegradation values, and their functionalities and environmental friendliness were also better than the unmodified DEP molecule.

### 3.3. Analysis of the Microbial Degradation Mechanism of DEP and Its Derivatives Based on a Microbial Degradation Path Simulation

#### 3.3.1. Simulation of Microbial Degradation of DEP and Its Derivative Molecules 

Plasticizer residue poses a serious threat to the environment. Among the possible ways of degradation, microbial degradation is the primary elimination process of PAE plasticizers. It achieves the purpose of recycling elements and balancing the ecosystem [45]. Gram-negative and Gram-positive bacteria degrade PAEs via different pathways, but both eventually form protocatechate [46]. Protocatechate can convert into pyruvate, succinate, and oxaloacetate, entering the tricarboxylic acid cycle and finally mineralizing into CO_2_ and H_2_O [47]. Ren et al. [48] found that esterase plays a crucial role in the microbial degradation of PAEs while studying their microbial degradation mechanism. Ester bond hydrolysis forms the key initial step in the microbial degradation of PAEs [49]. Based on the microbial degradation path of PAEs, the screened DEP derivative molecules (DEP-27, DEP-28 and DEP-29) were taken as examples to simulate and derive the microbial degradation path before and after DEP molecular modification (Figure 4). A Gaussian calculation was carried out for the energy barrier of the reaction to compare the difficulty of the biodegradation process before and after the DEP molecular modification (Table 6).

As shown in Figure 4, the microbial degradation transformation paths of DEP and its derivatives are divided into the transformation paths of Gram-negative bacteria and Gram-positive bacteria. Under the action of the plasticizer-degrading enzymes, DEP first hydrolyzed to phthalate monoesters (M0-1) and then hydrolyzed to form phthalate (M0-2). Phthalate 4,5-dioxygenase in Gram-negative bacteria oxidized the hydroxyl phthalate to produce 4,5-dihydroxyphthalic acid (M0-3), which decarboxylated to protocatechate (M0-5) [50]. Gram-positive bacteria hydrolyzed phthalic acid at C_3_ and C_4_ to produce 3, 4-dihydroxyphthalic acid (M0-4), which decarboxylated to protocatechate (M0-5) [51]. Finally, protocatechate mineralized into CO_2_ and H_2_O through the tricarboxylic acid cycle, culminating in the complete degradation of PAEs [47]. When the C_2_ site of DEP was replaced with -CONH_2_, the resulting derivative was difficult to hydrolyze into phthalic acid [52]. So the degradation process was slightly different from that of DEP.

#### 3.3.2. Calculation of the Reaction Energy Barrier for Microbial Degradation Transformation Paths of DEP and Its Derivatives

The reaction energy barrier required for the microbial degradation of the three DEP derivatives was lower than that required for the degradation of DEP (Table 6). A smaller reaction barrier is desirable as it improves the likelihood of the reaction path to occur. This indicates that the microorganisms could easily degrade the three DEP derivatives compared with DEP. The change rates of the reaction energy barrier for DEP-27, DEP-28, and DEP-29 were −20.15%, −23.42%, and −30.26%, respectively. These were consistent with the enhanced comprehensive biodegradability predicted by the comprehensive biodegradability model (DEP-27: 23.33%, DEP-28: 27.04% and DEP-29: 31.85%). This validates the reliability of the comprehensive biodegradation model, and indicates that the biodegradability of DEP can improve by molecular modification.

#### 3.3.3. Simulation and Verification of the Molecular Dynamics of the Microbial Degradation of DEP and Its Derivatives

In this paper, GROMACS 4.6.5 software was used to simulate the molecular dynamics (MD) of the ligand complex structures of DEP and its derivatives in Sybyl-x2.0, docked with phthalate dioxygenase reductase, esterase and acid esterase. The Poisson–Boltzmann surface area (MM-PBSA) method was used to calculate the binding free energy in the molecular dynamics [53]. Compared with the scoring function of molecular docking, the degree of binding energy characterized the biodegradability of DEP molecules before and after modification (Table 7).

Higher docking scores, indicate a stronger binding of the molecules to the degradative enzyme [54]. Additionally, the binding energy is directly proportional to the affinity between the molecule and the degrading enzyme [55], indicating a higher biodegradability of the molecule. Compared with the DEP molecules, DEP-27 and DEP-28 had lower scoring functions and higher free energy of binding to the degrading enzymes (Table 7). The scoring functions of only DEP-29 binding to the three degrading enzymes increased significantly, and its binding free energy also decreased significantly. The results demonstrated that DEP-29 was the only derivative that could easily bind to the three degrading enzymes at the same time and that the degree of binding was significant. This not only proves the rationality of the comprehensive biodegradation model of PAEs to modify the derivatives, but also forms the means to screen the PAE derivatives that synergistically degrade with microplastics.

## 4. Conclusions

In this paper, a 3D-QSAR model of the comprehensive biodegradability of the phthalic acid esters (PAEs) was constructed by combining the range normalization and entropy weight methods. In combination with molecular modifications, this model was successfully applied to design environmentally friendly PAEs that can co-degrade with microplastics. These PAE derivatives significantly improve the biodegradability of PAE plasticizers in microplastics, and can reduce the residual PAEs in the natural environment, relieve the adverse effects of plasticizer residues in the human body and environment, and provide the directional selection and theoretical support for the replacement of plasticizers. Further research work will be focused on whether the designed PAE derivatives could be degraded synergistically with the plasticizer-degrading bacteria or not, and the synthesis and biodegradation of these derivatives would provide an effective verification for the molecular modification method available in this study.

## Figures and Tables

**Figure 1 ijerph-17-05299-f001:**
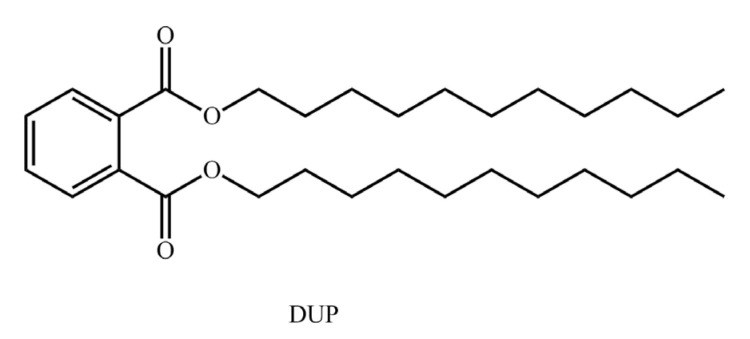
Molecular structure of the template molecule diundecyl phthalate (DUP).

**Figure 2 ijerph-17-05299-f002:**
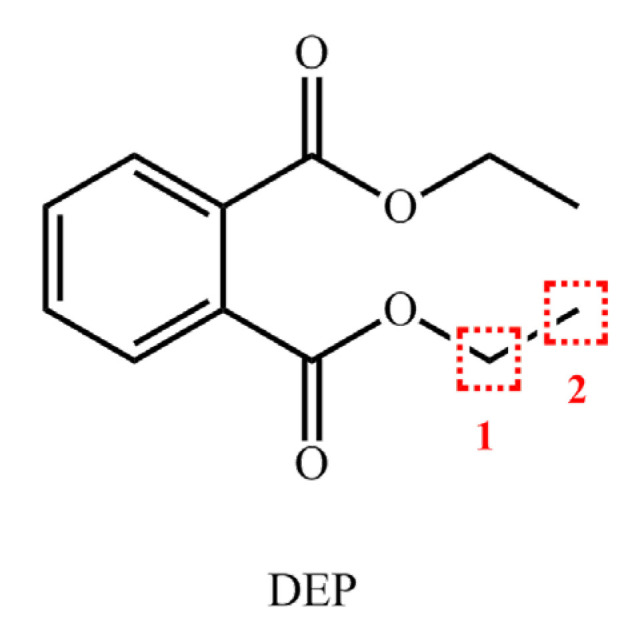
Molecular structure of the target molecule diethyl phthalate (DEP).

**Figure 3 ijerph-17-05299-f003:**
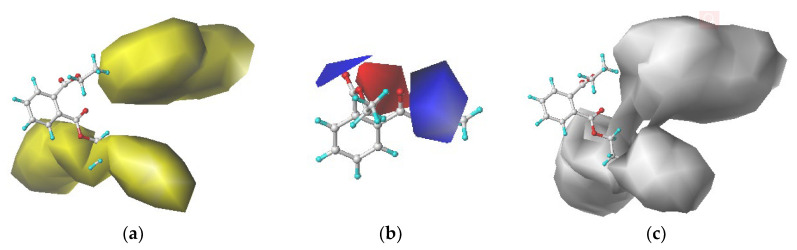
The contour maps from the CoMSIA model of DEP, (**a**) steric field, (**b**) electrostatic field and (**c**) hydrophobic field.

**Figure 4 ijerph-17-05299-f004:**
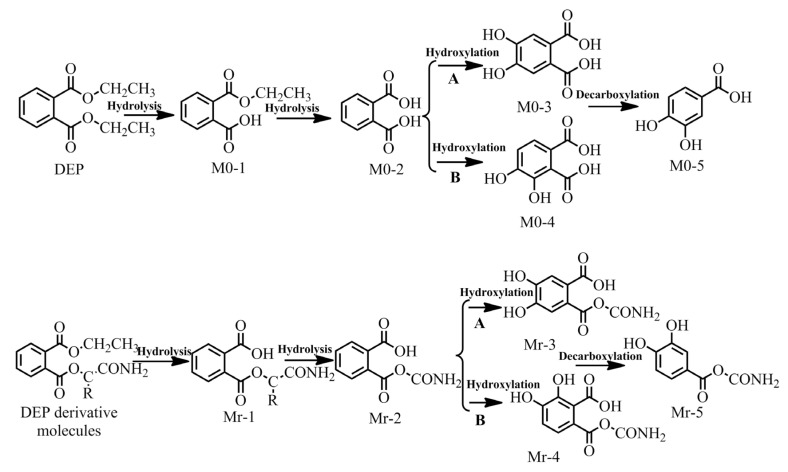
Schematic diagram of transformation paths of microbial degradation of DEP and its derivative molecules. A: The transformation path of gram-negative bacteria; B: the transformation path of Gram-positive bacteria; M0: the microbial degradation products of DEP; Mr: the microbial degradation products of DEP derivative molecules; R: -CH_3_, -CH_2_CH_3_, -CH (CH_3_)_2_; r = 1, 2, 3.

**Table 1 ijerph-17-05299-t001:** Biodegradability score values and comprehensive biodegradability values of the phthalic acid ester (PAE) molecules by three plasticizer-degrading bacteria.

Compounds	Docking Score Value of 2PIA	Converted Values of 2PIA	Docking Score Value of 2ZYI	Converted Values of 2ZYI	Docking Score Value of 3CN7	Converted Values of 3CN7	Comprehensive Biodegradation Values
BBP ^a^	7.070	0.509	1.908	0.100	5.116	0.380	0.323
DAP ^a^	5.918	0.385	5.461	0.357	5.275	0.409	0.380
DBP ^b^	5.754	0.367	5.551	0.364	6.047	0.547	0.408
DEP ^a^	5.574	0.347	4.711	0.303	3.548	0.100	0.272
DHP ^a^	7.455	0.551	7.221	0.485	6.964	0.711	0.564
DIBP ^a^	4.824	0.266	5.081	0.330	5.006	0.361	0.313
DIHP ^a^	8.634	0.679	10.570	0.727	7.870	0.872	0.743
DIHXP	7.462	0.552	10.548	0.726	6.660	0.656	0.643
DIPP ^a^	6.313	0.427	5.916	0.390	6.061	0.549	0.442
DIPRP ^a^	5.339	0.322	4.609	0.296	5.885	0.518	0.358
DMEP ^b^	7.085	0.511	4.088	0.258	6.876	0.695	0.459
DMP ^a^	3.293	0.100	4.624	0.297	3.940	0.170	0.191
DNOP ^a^	10.420	0.873	4.797	0.309	8.333	0.955	0.679
DPP ^a^	5.484	0.338	4.970	0.322	6.313	0.594	0.393
DPRP ^b^	6.273	0.423	6.681	0.446	4.746	0.314	0.406
DTDP ^a^	9.112	0.731	13.722	0.956	8.584	1.000	0.880
DUP *	11.595	1.000	14.334	1.000	8.561	0.996	0.999
*Ej*	1.799		1.792		1.497		−
*H_j_*	0.799		0.792		0.497		−
*W_j_*	38.28%		37.93%		23.79%		−

^a^: Training set; ^b^: test set; *: template molecule.

**Table 2 ijerph-17-05299-t002:** Evaluation parameters of the comparative molecular similarity index analysis (CoMSIA) model for the biodegradability of the PAEs by three plasticizer-degrading bacteria.

Model	q^2^	*n*	SEE	R^2^	F	r^2^_pred_	Q^2^	cSDEP	dq^2^/dr^2^yy	S	E	H	D	A
CoMSIA	0.731	8	0.013	0.999	578.738	0.761	0.385	0.251	1.488	34.7%	11.1%	46.8%	7.4%	0.0%

**Table 3 ijerph-17-05299-t003:** Prediction of the CoMSIA model of comprehensive and the single biodegradability of DEP derivatives and their change ratios.

No.	Substituent Group	Comprehensive Biodegradation Values	Change Rate (%)	Docking Score Value of 2PIA	Change Rate (%)	Docking Score Value of 2ZYI	Change Rate (%)	Docking Score Value of 3CN7	Change Rate (%)	Ratio
DEP		0.27		5.57		4.71		3.55		38.28:37.93:23.79
DEP-1	H_1_-CH_3_	0.293	8.52%	5.637	1.20%	5.232	11.08%	4.319	21.66%	−
DEP-2	H_1_-CH_2_CH_3_	0.311	15.19%	5.762	3.45%	5.353	13.65%	4.329	21.94%	−
DEP-3	H_1_-CH(CH_3_)_2_	0.328	21.48%	5.876	5.49%	5.397	14.59%	4.399	23.92%	−
DEP-4	H_1_-C(CH_3_)_3_	0.349	29.26%	5.976	7.29%	5.673	20.45%	4.36	22.82%	−
DEP-5	C_1_-CH_3_	0.269	−0.37%	5.500	−1.26%	4.827	2.48%	4.262	20.06%	−
DEP-6	C_1_-OH	0.228	−15.56%	4.828	−13.32%	4.838	2.72%	4.224	18.99%	−
DEP-7	C_1_-H	0.272	0.74%	5.059	−9.17%	4.784	1.57%	4.372	23.15%	−
DEP-8	C_1_-CN	0.255	−5.56%	5.266	−5.46%	4.874	3.48%	4.172	17.52%	−
DEP-9	C_1_-NH_2_	0.225	−16.67%	4.967	−10.83%	4.677	−0.70%	4.327	21.89%	−
DEP-10	C_1_-CHO	0.255	−5.56%	4.938	−11.35%	5.077	7.79%	4.113	15.86%	−
DEP-11	C_2_-OH	0.249	−7.78%	5.658	1.58%	4.594	−2.46%	4.29	20.85%	−
DEP-12	C_2_-CHO	0.262	−2.96%	5.737	3.00%	4.871	3.42%	4.282	20.62%	−
DEP-13	C_2_-COOH	0.269	−0.37%	5.892	5.78%	4.921	4.48%	4.312	21.46%	−
DEP-14	C_2_-NH_2_	0.256	−5.19%	5.504	−1.18%	4.911	4.27%	4.199	18.28%	−
DEP-15	C_2_-COCH_3_	0.289	7.04%	6.109	9.68%	4.969	5.50%	4.378	23.32%	−
DEP-16	C_2_-CONH_2_	0.235	−12.96%	5.755	3.32%	4.776	1.40%	4.436	24.96%	−
DEP-17	C_2_-(OH)_2_	0.267	−1.11%	5.484	−1.54%	5.495	16.67%	4.170	17.46%	−
DEP-18	C_2_-(CHO)_2_	0.267	−1.11%	5.409	−2.89%	5.73	21.66%	4.131	16.37%	−
DEP-19	C_2_-(COOH)_2_	0.284	5.19%	5.621	0.92%	5.826	23.69%	4.119	16.03%	−
DEP-20	C_2_-(NH_2_)_2_	0.259	−4.07%	5.444	−2.26%	5.434	15.37%	4.295	20.99%	−
DEP-21	C_2_-(COCH_3_)_2_	0.310	14.81%	5.799	4.11%	6.015	27.71%	4.188	17.97%	−
DEP-22	C_2_-(CONH_2_)_2_	0.261	−3.33%	5.362	−3.73%	5.946	26.24%	4.175	17.61%	−
DEP-23	H_1_-CH_3_-C_2_-COCH_3_	0.357	32.22%	6.317	13.41%	5.547	17.77%	4.198	18.25%	27.13:35.95:36.92
DEP-24	H_1_-CH_2_CH_3_-C_2_-COCH_3_	0.377	39.63%	6.453	15.85%	5.725	21.55%	4.169	17.44%	28.91:39.30:31.80
DEP-25	H_1_-CH(CH_3_)_2_-C_2_-COCH_3_	0.389	44.07%	6.565	17.86%	5.708	21.19%	4.200	18.31%	31.14:36.94:31.92
DEP-26	H_1_-C(CH_3_)_3_-C_2_-COCH_3_	0.406	50.37%	6.725	20.74%	5.787	22.87%	4.282	20.62%	32.29:35.60:32.11
DEP-27	H_1_-CH_3_-C_2_-CONH_2_	0.333	23.33%	6.077	9.10%	5.778	22.68%	4.129	16.31%	18.93:47.15:33.92
DEP-28	H_1_-CH_2_CH_3_-C_2_-CONH_2_	0.343	27.04%	6.151	10.43%	5.898	25.22%	4.130	16.34%	20.06:48.51:31.42
DEP-29	H_1_-CH(CH_3_)_2_-C_2_-CONH_2_	0.356	31.85%	6.263	12.44%	5.879	24.82%	4.144	16.73%	23.04:45.97:30.99
DEP-30	H_1_-C(CH_3_)_3_-C_2_-CONH_2_	0.370	37.04%	6.382	14.58%	5.953	26.39%	4.230	19.15%	24.25:43.89:31.86

Note: H_1_ stands for the H atom on the C_1_ site.

**Table 4 ijerph-17-05299-t004:** Evaluation parameters of *B. cepacia* (a), *A. fulgidus* (b), *P. aeruginosa* (c). Biodegradation 3D-QSAR model of PAEs.

Model	3D-QSAR	q^2^	n	SEE	R^2^	F	r^2^_pred_	Q^2^	cSDEP	dq^2^/dr^2^yy
a	CoMSIA	0.627	8	0.172	0.998	284.548	0.657	0.559	2.78	0.958
b	CoMFA	0.697	3	0.518	0.986	207.862	0.918	0.422	3.306	1.372
c	CoMFA	0.68	10	0.001	1	465107.312	0.618	0.491	3.021	0.690

**Table 5 ijerph-17-05299-t005:** Prediction parameters for DEP’s molecular function and environmental friendliness before and after modification.

No.	Total Energy (a.u.)	Change Rate (%)	Energy Gap (eV)	Frequency (cm^−1^)	Bio-Toxicity (logLC_50_)	Change Rate (%)	Bioaccumulation (logBCF)	BCF	Persistence (logt_1/2_)	Change Rate (%)	Long-Range Migration (log*K_OA_*)	Change Rate (%)
DEP	−766.62		5.32	24.02	1.100		1.264	18.37	3.156		7.505	
DEP-23	−919.26	−19.91%	5.15	15.09	0.781	29.00%	1.879	75.68	3.271	−3.64%	8.540	13.79%
DEP-24	−958.57	−25.04%	5.16	18.09	0.744	32.36%	1.992	98.17	3.250	−2.98%	8.546	13.87%
DEP-25	−997.89	−30.17%	5.14	17.65	0.737	33.00%	2.041	109.90	3.230	−2.34%	8.563	14.10%
DEP-26	−1037.2	−35.30%	5.06	15.38	0.672	38.91%	2.116	130.62	3.215	−1.87%	8.766	16.80%
DEP-27	−935.32	−22.01%	5.00	19.32	1.070	2.73%	1.569	37.07	3.386	−7.29%	8.329	10.98%
DEP-28	−974.64	−27.13%	4.97	16.33	1.047	4.82%	1.600	39.81	3.378	−7.03%	8.352	11.29%
DEP-29	−1013.95	−32.26%	4.96	16.24	1.038	5.64%	1.653	44.98	3.357	−6.37%	8.374	11.58%
DEP-30	−1053.26	−37.39%	4.94	15.99	0.923	16.09%	1.716	52.00	3.349	−6.12%	8.489	13.11%

**Table 6 ijerph-17-05299-t006:** Calculation of the reaction energy barrier of microbial degradation transformation paths of DEP and its derivatives.

DEP					Change Rate (%)	DEP-27				Change Rate (%)
Path	Reactants	Reaction Products	Energy Barrier (kJ/mol)	Total Energy Barrier (kJ/mol)		Reactants	Reaction Products	Energy Barrier (kJ/mol)	Total Energy Barrier (kJ/mol)	
Path1	DEP	M0-1	27.57	127.33	−	DEP-27	M1-1	28.88	113.16	−11.13
	M0-1	M0-2	15.75			M1-1	M1-2	13.92		
	M0-2	M0-3	53.82			M1-2	M1-3	56.97		
	M0-3	M0-5	30.19			M1-3	M1-5	13.39		
Path2	DEP	M0-1	27.57	139.68	−	DEP-27	M1-1	28.88	127.07	−9.02
	M0-1	M0-2	15.75			M1-1	M1-2	13.92		
	M0-2	M0-4	68.79			M1-2	M1-4	72.99		
	M0-4	M0-5	27.57			M1-4	M1-5	11.29		
						**Total Change rate** (%)				−20.15
**DEP-28**					**Change Rate** **(%)**	**DEP-29**				**Change Rate** **(%)**
**Path**	**Reactants**	**Reaction** **Products**	**Energy Barrier (kJ/mol)**	**Total Energy Barrier (kJ/mol)**		**Path**	**Reactants**	**Reaction** **Products**	**Energy Barrier (kJ/mol)**	
Path1	DEP-28	M2-1	24.94	110.98	−12.84	DEP-29	M3-1	17.07	99.77	−21.65
	M2-1	M2-2	15.67			M3-1	M3-2	12.34		
	M2-2	M2-3	56.97			M3-2	M3-3	56.97		
	M2-3	M2-5	13.39			M3-3	M3-5	13.39		
Path2	DEP-28	M2-1	24.94	124.90	−10.58	DEP-29	M3-1	17.07	113.68	−18.61
	M2-1	M2-2	15.67			M3-1	M3-2	12.34		
	M2-2	M2-4	72.99			M3-2	M3-4	72.99		
	M2-4	M2-5	11.29			M3-4	M3-5	11.29		
**Total Change rate** (%)					−23.42	**Total Change rate** (%)				−30.26

**Table 7 ijerph-17-05299-t007:** Molecular docking scores and molecular dynamics simulation of the binding energy calculation for DEP and its derivative molecules.

No.	2PIA		2ZYI		3CN7	
	Docking Score Value	△G_bind_ (kJ/mol)	Docking Score Value	△G_bind_ (kJ/mol)	Docking Score Value	△G_bind_ (kJ/mol)
DEP	5.574	−62.400	4.711	−138.694	3.548	−108.742
DEP-27	5.323↓	−126.613↓	3.491↓	−158.330↓	5.661↑	−102.247↑
DEP-28	5.793↑	−74.505↓	3.993↓	−138.588↑	6.139↑	−136.861↓
DEP-29	5.717↑	−108.149↓	7.535↑	−177.961↓	7.042↑	−160.312↓

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
