# Peer review of "Enhanced Biodegradation of Phthalic Acid Esters’ Derivatives by Plasticizer-Degrading Bacteria (Burkholderia cepacia, Archaeoglobus fulgidus, Pseudomonas aeruginosa) Using a Correction 3D-QSAR Model"

_ijerph, 2020, doi:10.3390/ijerph17155299_

Round 1
Reviewer 1 Report
The authors present a method for developing PAEs derivatives that may be more suitably degraded microbially with three commonly found enzymes using a 3D-QSAR model, molecular docking, and CoMSIA. Overall, the methodology as presented is clearly stated and the results are systematically outlined in the manuscript. It would be better for the authors to more clearly define microplastics and differentiate them from the focus of their study, the PAEs. The title, for instance, is a bit deceiving suggesting that the study focuses on synergistic degradation of PAEs and microplastics, when in fact the focus is on deriving simulations of potential degradabilities of different PAE derivatives, specifically DEP, using enzymes associated with the three bacteria listed. While PAEs are components of plastics, they are not microplastics and degradability of microplastics is not addressed in this study. A clear definition of what constitutes microplastics is needed and to imply that any synergism in degradation would occur is merely speculative. Having said that, the paper does provide a clear modeling approach for simulating the development and testing (via the software) of potential PAE derivatives to, as the authors state "provide directional selection and theoretical support for the replacement of plasticizers."
Since there is no line numbering, specific comments or edits will be provided as best possible:
p2L5: 'respiration' should likely be 'inhalation'
p2L5: "causing great harm to human health"
p2L5: "ecological environment safety" > choose one either ecological or environmental safety
p2 Para. 2 L3-5: "They observed that only specific microorganisms could grow on the surface of microplastics, thus affecting migration behavior and biodegradability of microplastics. During microbial degradation, the PAEs could synergistically degrade with the microplastics."
Growing on the surface doesn't necessarily mean that the organisms are consuming the plastics. Depending on the polymer type, degradability may be very low, and since many plastics have anti-fungal, UV protectant chemicals, the surface may only be acting as a place to live. I am also not sure about the synergistically degrading with microplastics. How is the degradation of PAEs and microplastics occurring synergistically?
p2 Para. 2 : "Several scientists attempted to isolate a single species of bacteria with a high-efficiency of degradation of the microplastics. Wu et al. [21] screened and isolated Ochrobactrum lupini and Agrobacterium tumefaciens that used dibutyl phthalate (DBP) as the only carbon source and energy source from the river sludge polluted with microplastics and completely mineralized it."
These sentences are somewhat deceptive in that the implication is that the microplastics are being completely degraded when in fact it is the PAEs that are being broken down. A better separation between microplastics and their constituents is needed in the manuscript.
p2 Para. 3: "In this paper, we employed the range normalization method combined with the entropy weight method to characterize the comprehensively biodegradability of the PAE by enzymes from three plasticizer-degrading bacteria."
I am not personally familiar with these combination of methods, especially the EWM, but from a quick review of available literature there are some general concerns with the method (see Zhu et al. 2020 [https://doi.org/10.1155/2020/3564835]). However, based on the fact that the indices used in the current study are based on specific properties of the PAEs, the method applied may be fine.
Otherwise, the rest of the manuscript presents work that is systematic and has good rationale supporting the results. A key detraction is that this is just a methodology and it will take some empirical testing to validate whether the simulations are in fact useful in a real life context. The authors could point to what the next steps would be for validation of their modeling of DEP to support the outcomes predicted.
Author Response
Responses to Reviewers' Comments
(Manuscript Number: IJERPH-864184)
The authors would like to thank the editor and reviewer for their careful reading of our manuscript and constructive comments and suggestions. We carefully considered the reviewer's comments or recommendations and addressed them point-by-point. All the revised sections have been highlighted in Yellow in the manuscript and the detailed responses to the comments are listed below.
Reviewer: 1
Comments to the Author
The authors present a method for developing PAEs derivatives that may be more suitably degraded microbially with three commonly found enzymes using a 3D-QSAR model, molecular docking, and CoMSIA. Overall, the methodology as presented is clearly stated and the results are systematically outlined in the manuscript. It would be better for the authors to more clearly define microplastics and differentiate them from the focus of their study, the PAEs. The title, for instance, is a bit deceiving suggesting that the study focuses on synergistic degradation of PAEs and microplastics, when in fact the focus is on deriving simulations of potential degradabilities of different PAE derivatives, specifically DEP, using enzymes associated with the three bacteria listed. While PAEs are components of plastics, they are not microplastics and degradability of microplastics is not addressed in this study. A clear definition of what constitutes microplastics is needed and to imply that any synergism in degradation would occur is merely speculative. Having said that, the paper does provide a clear modeling approach for simulating the development and testing (via the software) of potential PAE derivatives to, as the authors state "provide directional selection and theoretical support for the replacement of plasticizers." Since there is no line numbering, specific comments or edits will be provided as best possible:
1.The title, for instance, is a bit deceiving suggesting that the study focuses on synergistic degradation of PAEs and microplastics, when in fact the focus is on deriving simulations of potential degradabilities of different PAE derivatives, specifically DEP, using enzymes associated with the three bacteria listed.
Response: Thank you very much for your comments and suggestions. The title “Enhanced biodegradation of PAEs’ derivatives synergistically with microplastics degrading-bacterium such as Burkholderia cepacia, Archaeoglobus fulgidus, Pseudomonas aeruginosa” has been changed to “Enhanced biodegradation of phthalic acid esters’ derivatives by plasticizer degrading bacteria (Burkholderia cepacia, Archaeoglobus fulgidus, Pseudomonas aeruginosa) using a correction 3D-QSAR model”.
2.p2L5: 'respiration' should likely be 'inhalation'
Response: Thank you very much for your comments and suggestions. The word “respiration” has been changed to “inhalation” in the p2L5.
3.p2L5: "causing great harm to human health"
Response: Thank you very much for your comments and suggestions. The sentence “causing great harm to the human health” has been changed to “causing great harm to human health”.
4.p2L5: "ecological environment safety" > choose one either ecological or environmental safety
Response: Thank you very much for your comments and suggestions. The phrase “ecological environment safety” has been changed to “environmental safety”.
5.p2 Para. 2 L 3-5: "They observed that only specific microorganisms could grow on the surface of microplastics, thus affecting migration behavior and biodegradability of microplastics. During microbial degradation, the PAEs could synergistically degrade with the microplastics."Growing on the surface doesn't necessarily mean that the organisms are consuming the plastics. Depending on the polymer type, degradability may be very low, and since many plastics have anti-fungal, UV protectant chemicals, the surface may only be acting as a place to live. I am also not sure about the synergistically degrading with microplastics. How is the degradation of PAEs and microplastics occurring synergistically?
Response: Thank you very much for your comments and suggestions. All descriptions related to “degraded synergistically with microplastics degrading-bacterium” in the text were changed into “degraded by the plasticizer degrading bacteria.” And the ambiguous sentences “They observed that only specific microorganisms could grow on the surface of microplastics, thus affecting migration behavior and biodegradability of microplastics. During microbial degradation, the PAEs could synergistically degrade with the microplastics” hab been changed to “They observed that only specific microorganisms could affect migration behavior and biodegradability of plasticizer. During microbial degradation, the PAEs could simultaneously degraded by a variety of plasticizer degrading bacteria”.
6.p2 Para. 2: "Several scientists attempted to isolate a single species of bacteria with a high-efficiency of degradation of the microplastics. Wu et al. [21] screened and isolated Ochrobactrum lupini and Agrobacterium tumefaciens that used dibutyl phthalate (DBP) as the only carbon source and energy source from the river sludge polluted with microplastics and completely mineralized it." These sentences are somewhat deceptive in that the implication is that the microplastics are being completely degraded when in fact it is the PAEs that are being broken down. A better separation between microplastics and their constituents is needed in the manuscript.
Response: Thank you very much for your comments and suggestions. All descriptions related to “degraded synergistically with microplastics degrading-bacterium” in the text were changed into “degraded by the plasticizer degrading bacteria.” And the ambiguous sentences “Several scientists attempted to isolate a single species of bacteria with a high-efficiency of degradation of the microplastics. Wu et al. [21] screened and isolated Ochrobactrum lupini and Agrobacterium tumefaciens that used dibutyl phthalate (DBP) as the only carbon source and energy source from the river sludge polluted with microplastics and completely mineralized it” have been changed to “To improve the microbiological degradation efficiency of plasticizers, several scientists attempted to isolate a single species of bacteria with a high-efficiency of degradation from nature. Wu et al. [21] screened and isolated Ochrobactrum lupini and Agrobacterium tumefaciens that used dibutyl phthalate (DBP) as the only carbon source and energy source from the polluted river sludge and completely mineralized it.”
The ambiguous sentences “Microplastics pose a serious threat to the environment. Among the possible ways of degradation, microbial degradation is the primary elimination process of PAE plasticizers in the co-degradation of microplastics” have been changed to “Plasticizer residue poses a serious threat to the environment. Among the possible ways of degradation, microbial degradation is the primary elimination process of PAE plasticizers” in the 3.3.1
The ambiguous sentence “This not only proves the rationality of the comprehensive biodegradation model of PAEs to modify the derivatives, but also forms the means to screen the PAE derivatives that synergistically degrade with microplastics” has been changed to “This not only proves the rationality of the comprehensive biodegradation model of PAEs to modify the derivatives, but also forms the means to screen the PAE derivatives that simultaneously degraded by a variety of plasticizer degrading bacteria” in the 3.3.3.
The ambiguous sentence “In combination with molecular modifications, this model was successfully applied to design environmentally friendly PAEs that can co-degrade with microplastics. These PAE derivatives significantly improve the biodegradability of PAE plasticizers in the microplastics, reduce the residual PAEs in the natural environment, reduce the adverse effects of plasticizers residues in the human body and environment, and provide directional selection and theoretical support for the replacement of plasticizers” has been changed to “In combination with molecular modifications, this model was successfully applied to design environmentally friendly PAEs that can be simultaneously degraded by a variety of plasticizer degrading bacteria. These PAE derivatives significantly improve the biodegradability of PAE plasticizers, reduce the residual PAEs in the natural environment, relieve the adverse effects of plasticizers residues in the human body and environment, and provide directional selection and theoretical support for the replacement of plasticizers” in the conclusion.
7.p2 Para. 3: "In this paper, we employed the range normalization method combined with the entropy weight method to characterize the comprehensively biodegradability of the PAE by enzymes from three plasticizer-degrading bacteria." I am not personally familiar with these combination of methods, especially the EWM, but from a quick review of available literature there are some general concerns with the method (see Zhu et al. 2020 [https://doi.org/10.1155/2020/3564835]). However, based on the fact that the indices used in the current study are based on specific properties of the PAEs, the method applied may be fine.
Response: Thank you very much for your comments and suggestions. We have cited the recent paper by Zhu et al. 2020 [https://doi.org/10.1155/2020/3564835] in the p2 Para. 3
[24] Zhu Y, Tian D, Yan F. Effectiveness of Entropy Weight Method in Decision-Making [J]. Mathematical Problems in Engineering, 2020, 2020(7):1-5.
8.Otherwise, the rest of the manuscript presents work that is systematic and has good rationale supporting the results. A key detraction is that this is just a methodology and it will take some empirical testing to validate whether the simulations are in fact useful in a real life context. The authors could point to what the next steps would be for validation of their modeling of DEP to support the outcomes predicted.
Response: Thank you very much for your comments and suggestions. Suggestions were added in Conclusion as follow: Further research work will be focused on whether the designed PAEs derivatives could be degraded synergistically with the plasticizer degrading bacteria or not, and synthetise and biodegradation of these derivatives would provide effective verification for the molecular modification method availably in this study.
We have carefully revised all comments in the manuscript, many thanks to the editor and reviewers again.
Best regards,
Haigang Zhang
Reviewer 2 Report
The article from zhang et al. assess the biodegradability of phthalic acid esters (PAE), which are common plasticizers, by plasticizer degrading bacteria in order to better understand which type of PAE would have less impact on the environment. This article addresses an interesting and relevant topic and the conclusions seem useful. However, it is really hard to understand for a scientist who is not specialist of the field. Some concept and methods should be better explained for the article to become more accessible for a broader scientific audience. For example, what is the entropy weight method ? what is the CoMSIA model ? Are those standard method and model ? There are also many acronymes that are not explained at their first appearance that render the text difficult to follow, notably PAE, POPs, PDB, DUP.
Specific comments :
In the title the acronym PAE is used for the first time without explanation.
The abstract is lacking an introductory sentence to put the article into context. The way the methods and model are used in the abstract is not comprehensible for someone who is not familiar with it.
In the abstract, the sentence ‘It was used to modify the design environmentally friendly PAEs derivatives which can be synergistically degraded with microplastics’ has a weird structure.
In the abstract, POPs is used the first time without explanations.
In table 1, what does a biodegradability score represent ? Are there any units ?
Author Response
Responses to Reviewers' Comments
(Manuscript Number: IJERPH-864184)
The authors would like to thank the editor and reviewer for their careful reading of our manuscript and constructive comments and suggestions. We carefully considered the reviewer's comments or recommendations and addressed them point-by-point. All the revised sections have been highlighted in Yellow in the manuscript and the detailed responses to the comments are listed below.
Reviewer: 2
Comments to the Author:
The article from zhang et al. assess the biodegradability of phthalic acid esters (PAE), which are common plasticizers, by plasticizer degrading bacteria in order to better understand which type of PAE would have less impact on the environment. This article addresses an interesting and relevant topic and the conclusions seem useful. However, it is really hard to understand for a scientist who is not specialist of the field.
1.Some concept and methods should be better explained for the article to become more accessible for a broader scientific audience. For example, what is the entropy weight method? What is the CoMSIA model? Are those standard method and model? There are also many acronymes that are not explained at their first appearance that render the text difficult to follow, notably PAE, POPs, PDB, DUP.
Response: Thank you very much for your comments and suggestions. We have carefully explained the concept and methods are as follows:
The concept “The entropy weight method (EWM) is an objective method for weight processing, which uses information entropy for processing the weight” has been added in the 2.2.
The explanation of acronyms “Comparative molecular similarity index analysis (CoMSIA) model”, “phthalic acid esters (PAEs)” and “Persistent Organic Pollutants (POPs)” have been added in the abstract. “Diundecyl phthalate (DUP)” has been added in the 2.2.
2.In the title the acronym PAE is used for the first time without explanation. Response: Thank you very much for your comments and suggestions. The word “PAE” has been changed to “phthalic acid ester”.
3.The abstract is lacking an introductory sentence to put the article into context. The way the methods and model are used in the abstract is not comprehensible for someone who is not familiar with it.
Response: Thank you very much for your comments and suggestions. In the abstract, the sentence “This article aimed to characterize the comprehensive biodegradability of the phthalic acid (PAEs) by the plasticizer degrading bacteria (Burkholderia cepacia, Archaeoglobus fulgidus, Pseudomonas aeruginosa). The range normalization method combined with the entropy weight method was employed to standardize and objectively assign the biodegradability of the PAEs. On this basis, a CoMSIA model for comprehensive biodegradation of the PAEs was constructed. It was used to modify the design environmentally friendly PAEs derivatives which can be synergistically degraded with microplastics.” has been replaced with “A PAEs (phthalic acids) comprehensive biodegradability 3D-QSAR model was established, to design environmental-friendly PAEs derivatives, which could be simultaneously degraded by the plasticizer degrading bacteria, such as Burkholderia cepacia, Archaeoglobus fulgidus, Pseudomonas aeruginosa.”
4.In the abstract, the sentence ‘It was used to modify the design environmentally friendly PAEs derivatives which can be synergistically degraded with microplastics’ has a weird structure.
Response: Thank you very much for your comments and suggestions. The sentence “It was used to modify the design environmentally friendly PAEs derivatives which can be synergistically degraded with microplastics” has been deleted.
5.In the abstract, POPs is used the first time without explanations.
Response: Thank you very much for your comments and suggestions. The explanation “Persistent Organic Pollutants (POPs)” has been added in the abstract.
6.In table 1, what does a biodegradability score represent? Are there any units? Response: Thank you very much for your comments and suggestions. In table 1, the “Biodegradability score values” represent the binding degree of PAE to the three biodegradable enzymes, such as “Docking Score value of 2PIA”, “Docking Score value of 2ZYI”and “Docking Score value of 3CN7”. These docking score values have no units.
We have carefully revised all comments in the manuscript, many thanks to the editor and reviewers again.
Best regards,
Haigang Zhang